# Genome-Wide Identification of the *YABBY* Gene Family in Seven Species of Magnoliids and Expression Analysis in *Litsea*

**DOI:** 10.3390/plants10010021

**Published:** 2020-12-24

**Authors:** Xuedie Liu, Xing-Yu Liao, Yu Zheng, Meng-Jia Zhu, Xia Yu, Yu-Ting Jiang, Di-Yang Zhang, Liang Ma, Xin-Yu Xu, Zhong-Jian Liu, Siren Lan

**Affiliations:** 1College of Forestry, Fujian Agriculture and Forestry University, Fuzhou 350002, China; xdliu@fafu.edu.cn (X.L.); lxy-c@fafu.edu.cn (X.-Y.L.); 3190422059@fafu.edu.cn (Y.Z.); 1200455008@fafu.edu.cn (M.-J.Z.); 1180428005@fafu.edu.cn (Y.-T.J.); 2Key Laboratory of National Forestry and Grassland Administration for Orchid Conservation and Utilization at Colleage of Landscape Architecture, Fujian Agriculture and Forestry University, Fuzhou 350002, China; xiiiayu@fafu.edu.cn (X.Y.); deanyful@gmail.com (D.-Y.Z.); fjmaliang@fafu.edu.cn (L.M.); 1191775044@fafu.edu.cn (X.-Y.X.); zjliu@fafu.edu.cn (Z.-J.L.)

**Keywords:** magnoliids, *YABBY* gene family, *L. cubeba*, expression patterns

## Abstract

The *YABBY* gene family, specific to seed plants, encodes a class of transcription factors in the lamina maintenance and development of lateral organs. Magnoliids are sisters to the clade-containing eudicots and monocots, which have rapidly diversified among the common ancestors of these three lineages. However, prior to this study, information on the function of the *YABBY* genes in magnoliids was extremely limited to the third major clades and the early diverging lineage of Mesangiospermae. In this study, the sum of 55 *YABBY* genes including five genes in *INO*, six in *CRC*, eight in *YAB2*, 22 in *YAB5*, and 14 in *FIL* clade were identified from seven magnoliid plants. Sequence analysis showed that all encoded *YABBY* protein sequences possess the highly conserved YABBY domain and C2C2 zinc-finger domain. Gene and protein structure analysis indicates that a certain number of exons were highly conserved and similar in the same class, and *YABBY* genes encode proteins of 71–392 amino acids and an open reading frame of 216–1179 bp in magnoliids. Additionally, the predicted molecular weight and isoelectric point of YABBY proteins in three species ranged from 7689.93 to 43578.13 and from 5.33 to 9.87, respectively. Meanwhile, the *YABBY* gene homolog expression of *Litsea* was detected at a temporal and spatial level during various developmental stages of leaf and reproductive tissues. This research could provide a brief overview of *YABBY* gene family evolution and its differential expression in magnoliids. Therefore, this comprehensive diversification analysis would provide a new insight into further understanding of the function of genes in seven magnoliids.

## 1. Introduction

The *YABBY* gene family, a small class of transcription factors, belongs to the zinc-finger superfamily. As a class of transcription factors, it plays an important role in vegetative and reproductive tissue development including stress response, leaf margin, and adaxial polarity establishment in flowering plants [1,2,3,4]. Unlike other transcription factor families, *YABBY* genes seemed to be absent in non-seed lineages of embryophyte plants [5]. Additionally, *YABBY* genes encode putative transcription factors that consist of a zinc-finger domain in the N-terminal and C-terminal YABBY domain, similar to a high-mobility group (HMG) box [6,7]. In *Arabidopsis thaliana*, the *YABBY* gene family has a distinct role, declared to six subclades, including *CRABS CLAW* (*CRC*), *INNER NO OUTER* (*INO*), *FILAMENTOUS FLOWER* (*FIL*), *YABBY3* (*YAB3*), *YABBY2* (*YAB2*), and *YABBY5* (*YAB5*) [8,9,10]. Generally, *CRC* and *INO* are considered the “reproductive-specific *YABBY* genes”, whereas *FIL*, *YAB2*, *YAB3*, and *YAB5* are referred to as the “vegetative *YABBY* genes” [10,11]. Accordingly, the “vegetative *YABBY* genes” could regulate the lateral organ polarity development, formation of margins, maturation of the leaf, and development and phyllotaxis of the shoot apical meristem and are detected both in leaf primordia and floral organs [12,13,14], whereas two other genes, *CRC* and *INO*, are restricted to developing carpels and the outer integument of the ovule, respectively. Besides regulating carpels polarity, *CRC* genes are also required and essential for nectary specification [14]. On the basis of these findings, the *YABBY* gene family facilitate the abaxial cell fate of lateral organs in *Arabidopsis* [12].

According to the previous results from core eudicots, it is suggested that the “vegetative *YABBY* genes” were related to several aspects of leaf growth across angiosperms, including the intensive expression in ectopic growths on the abaxial and adaxial surfaces of polarity mutants, the construction of leaf polarity, and leaf margin establishment that guides lamina growth and leaf initiation [9,15]. For instance, previous research on *Antirrhinum majus* has indicated that *FIL*-like gene *GRAMINIFOLIA (GRAM)* and *YAB5* homologous gene *PROLONGATA (PROL)* were both expressed abaxially and promote laminar outgrowths, and *GRAM* could also regulate flower organ initiation and identity [16].

Studies in monocots focusing primarily on grass species have implied that varied expression patterns would lead to the divergence of proposed functions [17]. For example, *ZYB9* and *ZYB14*, belonging to the FIL/YAB3 clade, are essential for flower development and regulate the lateral outgrowth in *Zea mays* [18]. *OsYABBY1*, a *YAB2* homologous gene from rice, was closely associated with putative precursor cells that generate the mestome sheath in a large vascular bundle, the abaxial sclerenchyma in the leaves, the several layers of the palea and lemma, and it was thus suggested that it determines the differentiation of some specific cell types in grasses and seems to be unrelated to regulating the polarity establishment of lateral organs [12,19,20,21]. It has been shown that, before the diversification of angiosperms, four gene duplication events occurred in the *YABBY* gene family, leading to gene neo-functionalization [1,2,16]. Further, expression patterns from the early diverging angiosperms lineages have shown large variation. For instance, *AmbF1*, a *YAB2*-like homolog from *Amborella trichopoda*, was expressed adaxially in all floral organs, leaves, and shoot apices [1,16]. In contrast to *Amborella*, the *YABBY* genes from *Cabomba caroliniana* (*CcYAB5*, *CcFIL*, *CcINO*, and *CcCRC*) and other eudicots are expressed abaxially [1,16].

In most eudicot primordia, *YABBY* genes were expressed dissymmetrically and restricted to abaxial regions, whereas expression has been measured as being adaxial-specific among the leaf primordia initiation stage in *Zea* [1,4,11]. In eudicots, expression of the *CRC* gene are restricted to nectaries and carpels; however, *DROOPING LEAF* (*DL*), an ortholog of *CRC* in monocots, is involved in the development of the leaf midrib and regulation of carpel identity by excluding B-class *MADS-box* gene expression from the flower fourth whorl [13,19,22]. The second reproductive *YABBY* gene, *INO*, in cooperation with *SUPERMAN* (*SUP*), is involved in the asymmetric growth of the outer ovule integument, which is another example of tissue-specific functions of the *YABBY* genes in addition to the *CRC* gene [8,23]. In addition, *Arabidopsis INO* expression is restricted to the outermost cell layer of the outer integument and promotes the outgrowth in the outer integument of the ovule [24,25]. Furthermore, abaxial expression of the *INO* ortholog has also been reported in eudicots and several representatives in the ANA grade angiosperms [20,24,25].

The flowering plants are the most diverse group of plants, and the basal flowering plant lineages were diverged at a very early stage in angiosperms’ evolution, followed by rapid diversification among the magnoliids, eudicots, and monocots [26,27]. Gene family comparisons in whole genomes across plant lineages would contribute to identifying critical events in the evolution of major flowering plants [27,28]. The magnoliids constitute a significant group of flowering plants but are much smaller than the monocots and eudicots. Thus, it possesses a unique phylogenetic position to better understand the evolution of extant flowering plants. Furthermore, with two cotyledons, the basal angiosperms and magnoliids are not monocot, and with pollen having a single pore, they are also not eudicots [26]. However, little is known about the regulatory function of the *YABBY* genes in magnoliids; for example, apparent deletion of the *INO* locus in *Annona squamosa* would create spontaneous seedless mutants, suggesting conservation of the role of a critical regulator of ovule development between eudicots and more ancient lineages of angiosperms, and *CpCRC* (*Chimonanthus praecox*) was expressed in all asymmetric above-ground organs in a polarity, indicating that this gene is involved in establishing dorsiventral polarity in all of these organs [29,30]. Moreover, the relationships among magnoliids, eudicots, and monocots have not been conclusively resolved. Recently, several whole genomes of magnoliids species, including three Lauracea species, *Cinnamomum micranthum*, *Litsea cubeba*, and *Phoebe bournei*, two Calycanthaceae species, *C. praecox* and *Chimonanthus salicifolius*; *Piper nigrum* (Piperaceae), as well as *Liriodendron chinense* (Magnoliaceae), have been sequenced [27,28,31,32,33,34,35]. The completed assembly of the whole genome of these magnoliids would provide a new opportunity for a systematic exploration of the *YABBY* gene family.

On the whole, this study aimed to identify, classify, and compare *YABBY* genes on a genome-wide scale between *L. cubeba* and other magnoliids; moreover, the expression patterns of eight putative *Litsea YABBY* genes were analyzed using RNA-seq data in various organs.

## 2. Results

### 2.1. Isolation and Sequence Analysis of YABBY Genes from Magnoliids

In previous reports, 6, 8, and 13 *YABBY* genes were identified in *Arabidopsis*, *Oryza sativa*, and *Zea mays*. In total, 55 and 56 *YABBY* genes were identified via HMMER analysis and BLASTP search, respectively (Hmmsearch: eight in the genome of *C. micranthum* and *L. cubeba,* seven in *P. bournei*, six in *L. chinense* and *C. salicifolius*, five in *C. praecox*, and 15 in *P. nigrum*; BLASTP: the individual number of *YABBY* gene family was the same as those by using Hmmsearch, with the exception of eight in *P. bournei*). To identify the *YABBY* genes accurately from magnoliids, the conserved sequence of YABBY domain (PF04690) generated from the Pfam protein database was used to search the magnoliids’ predicted proteomes. Finally, eight members of the *YABBY* gene family were both identified in the genome of *C. micranthum* and *L. cubeba,* seven in that of *P. bournei*, six in that of *L. chinense* and *C. salicifolius*, five in that of *C. praecox*, and 15 in that of *P. nigrum*. Based on sequences verified by domain identification and the classification scheme of the *YABBY* gene family in other plants, 55 magnoliids *YABBY* genes were separated into five clades, including CRC-like, INO-like, YAB2-like, YAB5-like, and FIL-like groups, respectively (Table 1). Four species as well as *Arabidopsis* and rice have similar *YABBY* gene members, as previously reported (Table 1), and two species of Calycanthaceae presented in this report have five or six *YABBY* gene members due to the fact of losing one copy in *FIL* or the lack of the *INO* gene. In contrast to other species, *P. nigrum* contained twice as many gene members as those in other species as a result of the apparent small-scale segmental duplications or tandem duplications in three clades (two in YAB2-class, four in FIL-class, and seven in YAB5-class) (Table 1). However, there were still noticeably fewer gene members than found in some dicotyledonous, such as *Gossypium raimondii* (38), which may be due to the large sizes of these plants’ genomes or the genome duplication events that occurred in their evolution [10]. As previously reported, the total number of *YABBY* genes increased with the plant evolution and genome duplications, suggesting that these genes may play a significant role in regulating plant growth and development.

### 2.2. Congruence of YABBY and Zinc-Finger Domains

The *YABBY* genes constitute a small transcription factor family characterized by two conserved domains, an N-terminal region C2C2 zinc-finger domain and a C-region YABBY domain [7]. To examine the structural features of 55 YABBY proteins, multiple sequence alignments analysis of magnoliids YABBY proteins were applied to generate sequence logos of both the C2C2 zinc-finger and YABBY domains in seven magnoliids and *Arabidopsis* (Figure 1). Moreover, the letter size in the logo shows the conserved degree of this amino acid at this site.

Next, we dissected the sequence features of all magnoliid YABBY proteins using the MEME suite. A motif study revealed differential as well as unique motifs between distinct clades and species. The initial search was accomplished by using 55 YABBY proteins with the search parameters set to 15 motifs of 11 to 50 amino acids in width (Figure 2). The motif study illustrated that some identified motifs seemed to be clade-specific, and further detailed analyses were conducted with various sequences from individual or sister clades. Motif occurrence and relative position within the amino acid sequence were matched in relation to the zinc-finger and YABBY domains. As previously predicted, the amino acid sequences of these motifs were highly conserved and specific, like the sequence homologs previously detected in YABBY proteins from *Arabidopsis*, *Oryza*, and *Zea*. The C2C2 zinc-finger domain mostly consisted of motifs 2; motifs 1, 3, and 4 corresponded to the YABBY domain. Besides the common motifs 1, 2, 3, and 4 harbored by most *YABBY* genes analyzed, several motifs existed privately in genes within the same subfamilies. For example, the *FIL*-like group was privately characterized by motif 5, 10, and 12, and the YAB5 class (motif 9) was distinguished from *YAB2* genes if motif 9 existed. However, the functions of these putative motifs were not precise due to the lack of homologs in protein motif databases. The conserved motif 13 found in YABBY proteins, except *P. nigrum*, were clade-specifically distributed, indicating that this motif may be related to the *CRC*/*DL* gene function. In contrast with others, motif 13 may evolve with a new role in the YAB5 class within *P. nigrum*. Besides that, the motif location map also clearly showed that the conserved domains were not complete in all genes in the CRC clade or the INO clade.

### 2.3. Phylogenetic Relationship and Gene Structure of YABBY Genes in Magnoliids

To comprehend the evolutionary relationships among these *YABBY* genes in magnoliids, we constructed a phylogenetic tree on the basis of the YABBY full-length protein sequences together with the first diverging angiosperm lineages including YABBY proteins from *Amborella*, *Cabomba*, and *Nymphaea colorata*. The phylogenetic tree indicates that the magnoliid YABBY proteins could be divided into five clades: YAB5, YAB2, CLC(CRC/DL), INO, and FIL (Figure 3). Additionally, our analyses imply that *YABBY* genes may derive from *YAB5* genes in the early angiosperms, and *FIL*-like genes are well isolated from their closest paralogue group, containing CRC clade, and the earlier-diverged INO clade, through a series of strongly supported branches. The evolutionary tree also shows at least two orthologs of the *FIL* gene in five magnoliids, while the copy numbers of *FIL* gene remained as a single copy in Calycanthaceae (*C. praecox* and *C. salicifolius*). In brief, six of the magnoliid YABBY proteins (LcCRC, CmCRC, PboCRC, ChsCRC, ChpCRC, and PnCRC) were classified in the CLC clade, although none in *Liriodendron* and *C. praecox*, five in the INO clade, eight in the YAB2 clade, 14 in the FIL clade, and 22 in the YAB5 clade (Figure 3).

The phylogenetic result displays that the majority of magnoliids contain one member of the CLC clade, but there are no CRC proteins in Liriodendron (Figure 3). To further investigate the evolutionary relationships in the YABBY subfamily of magnoliids, CLC protein sequences from different angiosperms and magnoliids were obtained from the public database for further analysis (Figure 4). The phylogenetic analysis of CRC proteins demonstrated that most magnoliids CRC proteins are located at the base of the clade containing monocot and eudicot CRC proteins, with the exception of the PnCRC protein, which was clustered with Nymphaeales (NcCRC and CcCRC), and demonstrated that a duplication event might occur at the most recent common ancestor of monocots and eudicots, suggesting a possible sub-functionalization or neo-functionalization of the *CRC/DL* gene after magnoliids are separated from basal angiosperms.

Similar to the CLC clade, we found that each magnoliid contains only one member in the INO clade, as do most basal angiosperms (Figure 5). The INO members of monocots, grouped independently, except for the basal angiosperms, were at the base of other INO subclades. It is assumed that this group may not have undergone expansion, and group members may perform a biologically conserved function.

The phylogenetic analysis suggested that YAB2 members in the Piperaceae at the base of the subclade included eudicots and other magnoliids (Figure 6), implying that *YAB2* genes in eudicots may be derived from magnoliids. Furthermore, all YAB2 proteins of magnoliids, sisters to eudicots, are clustered into an independent group, except for *P. nigrum*, as they have a similar phylogenetic analysis to the YAB5 subfamily.

It is worth noting that *FIL*-like genes may have experienced gene expansions in monocots, magnoliids, and eudicots (Figure 7). Interestingly, we found that five species have more than two copies of *FIL*-like genes, while two from *Chimonanthus* have a single copy, similar to basal angiosperms (*AmtFIL*, *BsFIL*, *CcFIL*, *NjFIL*, and *NcFIL*), suggesting that the other gene may be absent in *Chimonanthus*. The most parsimonious explanation of this observation was that these two *FIL*-like genes were originated by the whole-genome duplication in the most recent common ancestor of magnoliids, eudicots, and monocots, while the phylogenetic tree was not clearly implied (Figure 7).

The YAB5 proteins of angiosperms are divided into four groups: eudicots, magnoliids, basal angiosperms, and *P. nigrum*. Interestingly, seven *YAB5*-like sequences from *P. nigrum* were clustered into one independent subclade based on a clade containing eudicots and magnoliids (Figure 8). The previous research results indicated that *YAB5*-like genes are eudicot-specific, since basal angiosperm sequences were ambiguous, and orthologues were not discovered from monocots. Contrary to the previous study, 22 *YAB5* genes were identified from seven magnoliids, which illustrates the first statement of *YAB5*-like genes for magnoliids (Figure 8). Moreover, the present data reveal that *YAB5* genes remain in the major angiosperm lineages, containing basal angiosperms, magnoliids, and eudicots, with monocots the only exception.

### 2.4. Gene Structure Analysis of YABBY Genes in Magnoliids

Exon/intron analysis could provide some valuable information regarding the evolutionary relationships among plant taxa [36]. Therefore, this study investigated and compared the exon/intron structures of *YABBY* genes among these seven magnoliids species. The previous reports explained that highly similar exon patterns were present in rice and *Cabomba YABBY* genes [16,21]. In this study, the results showed that most *YABBY* genes have more than one exon: 6 to 15 exons in CRC/DL, 2 to 7 in INO, 6 to 7 in YAB2, 2 to 9 in YAB5, and 7 to 12 in FIL clades (Table 2). Furthermore, the fourth and fifth exons of most genes have a similar length (exon 1: 68 bp; exon 4: 48 bp), and a similar exon length exists in the same clade; for instance, exon 2 (74 bp/77 bp) in *FIL*-like genes or exon 5 (75 bp) in *YAB5*-like genes. In addition, protein structure analysis also showed that *YABBY* genes in magnoliids encoded proteins of 71–392 amino acids and an ORF (open reading frame) of 216–1179 bp. The predicted molecular weight and isoelectric point of YABBY proteins in the three species ranged from 7689.93 to 43578.13 and from 5.33 to 9.87, respectively. In addition, the physical map positions of the *YABBY* gene family on chromosomes of *Arabidopsis*, *Oryza*, *Zea*, and magnoliids have been provided in the Appendix A.

### 2.5. Gene Expression Analysis in Litsea

The preceding studies indicated that *YABBY* genes are abaxially and conservatively expressed in seed plants [12]. To verify the expression patterns of the *YABBY* genes in magnoliids, we analyzed the abundance of different periods and parts from *Litsea* RNA-seq data, including the fruit, leaf, root, stem, and flower. The results show that the *FIL*-like gene of *Litsea* was mainly expressed at the abaxial leaf tissues, and *LcFIL.1* was much higher than that of *LcFIL* in this tissue (Figure 9), suggesting that the *LcFIL.1* gene might play a greater role than the *LcFIL* gene. *Litsea* contains three genes, *LcYAB5*, *LcYAB5.1*, and *LcYAB5.2*, in the YAB5 clade. On the one hand, the expression of *LcYAB5* genes was obviously extended to all periods of the leaf and stem organs, with moderate expression in the fruit and flower. On the other hand, *LcYAB2* transcripts could be significantly measured at the fruit as well as the leaf and stem. In this study, *Litsea* has only one member in the CRC and INO clades; the former was not expressed in all samples, and the latter showed low expression. These results suggest that members in these two clades may have a specific function in tissues, and members of the CRC clade might play a minor role in the growth and development of vegetative tissues in *Litsea*. This result implies that *LcINO* and *LcCRC* were expressed in the outer integument-specifically and carpel-specifically, respectively.

## 3. Discussion

*YABBY* genes are small and essential key regulators that are specifically related to the evolution of leaves in seed plants that encode a C2C2 zinc-finger domain at the N-terminal region and a conserved YABBY domain in the C-terminal region with a helix–loop–helix motif [37,38]. In terms of basal angiosperms, five *YABBY* genes have been identified in *Amborella*, *Cabomba,* and *Nymphaea* [36]. In addition, six *YABBY* genes have been described in the model plant *Arabidopsis*, and eight have been identified in *Oryza* [20,21,22,37]. The *YABBY* gene family plays an essential role in shoot meristem identity and regulating lateral organ development, including inflorescence, cotyledons, and leaf [9]. In this research, five to eight *YABBY* transcript factors were found from six magnoliids through a genome-wide search, and 15 were found in *P. nigrum*. Furthermore, the total number of *YABBY* genes in each magnoliid genome was distinguished from those analyzed in the monocot and dicot plants. The phylogenetic tree showed that the *YABBY* gene family in magnoliids, different from monocots absented in the YAB5 class [1], could be separated into five groups (YAB2, YAB5, FIL/YAB3, CRC and INO), and these were previously studied in dicotyledonous plants. It is clear that the YAB5 clade was exclusively composed of basal angiosperms and eudicots [39].

The group of ancestral angiosperms did not form a monophyletic group, but their lineages represent early splits from eudicots and monocots, following a divergence of flowering plants from common ancestors [26]. In this study, each magnoliid examined had a single copy in the INO, CRC, and YAB2 clades with the exception of Piper.

Previous studies have provided evidence that the expression signal of *Arabidopsis FIL*, *YAB2*, and *YAB5* genes have been detected in leaves, cotyledons, and floral organs and redundantly control the growth of lateral organs [12,15,40]. *TOB1*-related genes, orthologs of *FIL* genes, have similarly acted in flower development in *Oryza* [37,41]. Our results also displayed that four magnoliids retained two *FIL*-like genes, with four duplications in *P. nigrum* and only one copy in *Chimonanthus*. Expression of *FIL* gene in vegetative leaves coincided with the acquisition of a role in lateral organs [38]. According to the total expression matrix, expression signal of *YAB2*-like genes could be both tracked within reproductive and vegetative tissues. Previous work provided evidence that *YAB2*-like genes function to promote laminar outgrowth and stamen morphologic evolution [17]. Apart from conserved *YAB2* regulation, the dynamic regulation and biological function of *INO*-like and *CRC*-like genes should be conserved in angiosperm plants as consistent as that in the previous findings that *INO* and *CRC* genes were involved in the seed formation and differentiation of the nectary in basal taxa, respectively [14,16,22,28,29].

## 4. Materials and Methods

### 4.1. Identification of the YABBY Gene Family in Magnoliids

Two methods were employed to comprehensively identify the maximum number of YABBY and zinc-finger domain-containing sequences in magnoliids. The first method (HMM analysis) utilized the HMMER 3.2.1 software package used to build hidden Markov model profiles from full Pfam alignment files with the YABBY domain seed file (PF04690; http://pfam.xfam.org/) at an E value of 10^−5^. These models were further performed to search the protein database and identify potential YABBY proteins. In the second method (BLASTP search), the well-characterized *A. thaliana* YABBY proteins were used as the query file, which blasted against the magnoliid protein database to obtain the matching sequences. Unique and non-redundant magnoliid *YABBY* gene family members for the following analysis were determined by motifs and removing redundant gene sequences. The genome data for four species in magnoliids were retrieved from the downloaded data (Bioproject: *C. micranthum*, PRJNA477266; *L. cubeba*, PRJNA562049; *P. bournei*, PRJCA002001; *L. chinense*, PRJNA418360; *C. praecox*, PRJNA600650; *C. salicifolius*, PRJNA602413; *P. nigrum*, PRJNA529758).

### 4.2. Conserved Domains and Motifs in YABBY Proteins

Multiple full-length *YABBY* protein sequences from the magnoliids and other plants were aligned by the MUSCLE program (built-in MEGA 7.0). To investigate the conservation of the YABBY domain and the C2C2 zinc-finger domain in magnoliids and *Arabidopsis* YABBY proteins, the online tool WEBLOGO was adopted, and multiple sequence alignment results were regarded as the input file [42]. Furthermore, MEME Suite Version 5.1.1 was applied to examine the conserved motifs among 55 magnoliid YABBY proteins with the search parameters set to 15 motifs of 6 to 50 amino acids in width. Motifs were expected to occur zero or one occurrence per sequence [43]. The motifs obtained from seven magnoliids were annotated with the SMART [44] or NCBI-S-MARTBLAST search programs.

### 4.3. Gene and Protein Structures Analysis

The exon positions and sizes of identified magnoliid *YABBY* genes were acquired and calculated from the gff3 files using a Perl script, and the online Gene Structure Display Server GSDS 2.0 [45] was applied to perform the gene structures of identified *YABBY* genes. These genes were further compared to assembled transcriptomic sequences. To further study the *YABBY* gene family of magnoliids in the future, protein physical information of *YABBY* genes in this paper, such as the isoelectric point and molecular weight (pI and Mw), can be considered [46].

### 4.4. Phylogenetic Analyses of YABBY Gene Family in Magnoliids

The eudicots and monocots of YABBY proteins were collected from the NCBI database, Phytozome, or PlnTFDB (http://plntfdb.bio.uni-potsdam.de/v3.0/). A total of *YABBY* genes were picked for the purpose of classifying the types of *YABBY* genes in magnoliids. For phylogenetic analysis, MUSCLE or ClustalW (build-in MEGA 7.0) was performed to align full-length YABBY amino acids sequences from several seed plants. The neighbor-joining (NJ) phylogenetic tree of the *YABBY* gene family was constructed using MEGA 7.0 [47] with a P-distance model and pairwise deletion option parameters, and bootstrap analysis was set as 1000 iterations. In addition, phylogenetic trees of the *YABBY* gene family used to further infer the relationships among these genes were constructed by protein sequences using the NJ method with MEGA 7.0. The phylogenetic tree was visualized with Figtree (Version 1.4.4).

### 4.5. Transcriptome Data Analysis and Gene Expression Heatmap

The raw RNA-seq data of *Litsea* could be downloaded and collected from the NCBI Sequence Read Archive (PRJNA562115; https://www.ncbi.nlm.nih.gov/bioproject/PRJNA562115). Bowtie2 (Version 2.2.5) was employed for aligning the clean reads to the reference coding gene set, and the expression files of *YABBY* genes in *Litsea* were calculated by RSEM (V1.2.12) software. The expression levels of the *YABBY* gene family were decided by extracting their respective data from the total expression matrix, visualized with TBtools software [48].

## 5. Conclusions

In this study, 55 *YABBY* genes in magnoliids were counted from seven species and analyzed systematically. Our research provides more conclusive evidence that *YABBY* genes from magnoliids could be diversified into five distinct groups, as in other eudicots and basal angiosperms, each with unique and distinct sequence characteristics outside of the conserved amino acid domains. The transcriptome expression file indicated that the *YABBY* gene family determined significant functions in the development and growth of lateral organs, flowers, and fruits. Phylogenetic analysis showed that the single orthologs of YAB2, CRC, and INO subgroups exist in magnoliids and that the YAB5 class possesses at least two copies in magnoliids. The *FIL* genes are present in different numbers depending on the plant family, as there are two in Lauraceae, one in Calycanthaceae, and four in Piperaceae. In addition, the total number of *YABBY* gene family was expanded to 15 in Piper, beyond those of the other magnoliids, possibly due to the fat of polyploidization or natural selection. In conclusion, our results would represent a comprehensive genome-wide research of the *YABBY* gene family in magnoliids, which opens the gate to further detailed exploration on the gene function and evolution of *YABBY* genes in magnoliids.

## Figures and Tables

**Figure 1 plants-10-00021-f001:**
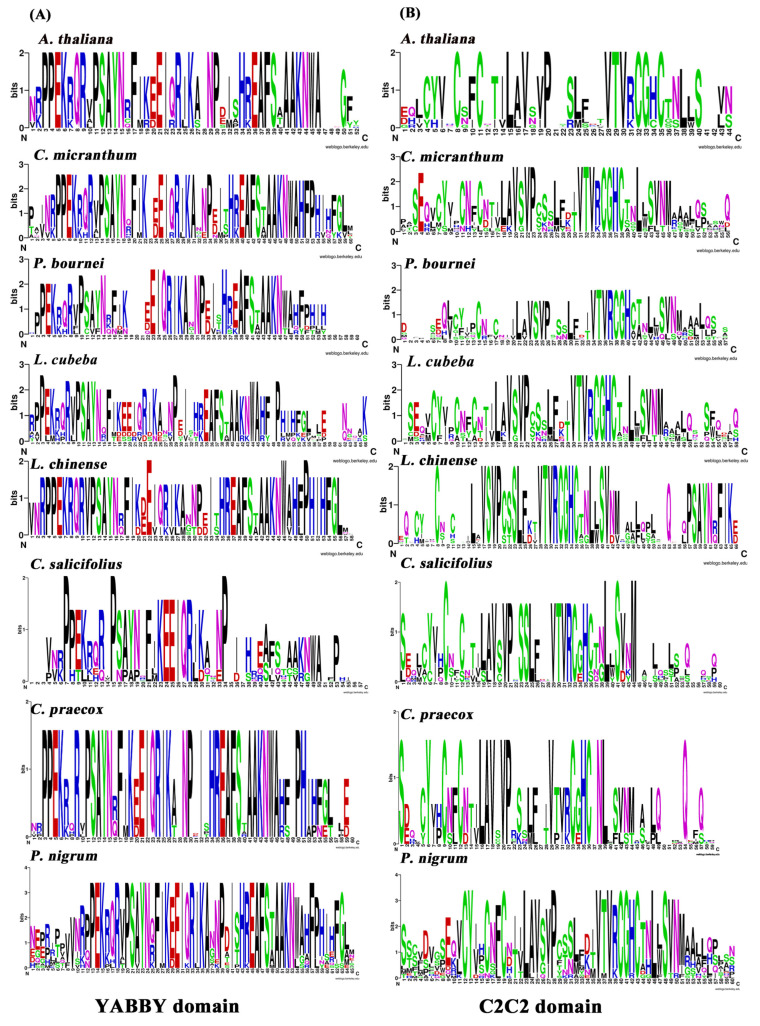
Conserved domain of seven magnoliids and *Arabidopsis thaliana*
*YABBY* protein sequences. (**A**) Sequence logo showing the C-terminal conserved YABBY domain. (**B**) Sequence logo revealing the N-terminal conserved zinc-finger domain.

**Figure 2 plants-10-00021-f002:**
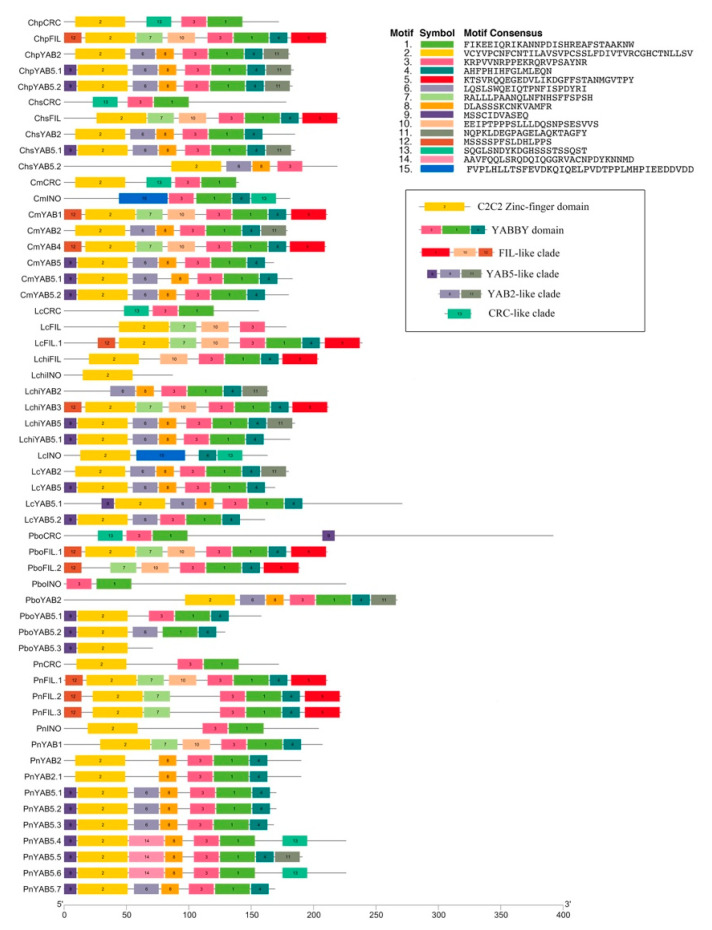
Conserved motifs analysis in YABBY proteins among seven magnoliids. Motifs were determined using MEME suite version 5.1.1. Black lines represent the non-conserved sequence, and each motif is indicated by a colored box numbered in the middle. The same acronym represents the same species: Cm: *C. micranthum*; Chp: *C. praecox*; Lc: *L. cubeba*; Chs: *C. salicifolius;* Pbo: *P. bournei*; Pn: *P. nigrum*; Lchi: *L. chinense*.

**Figure 3 plants-10-00021-f003:**
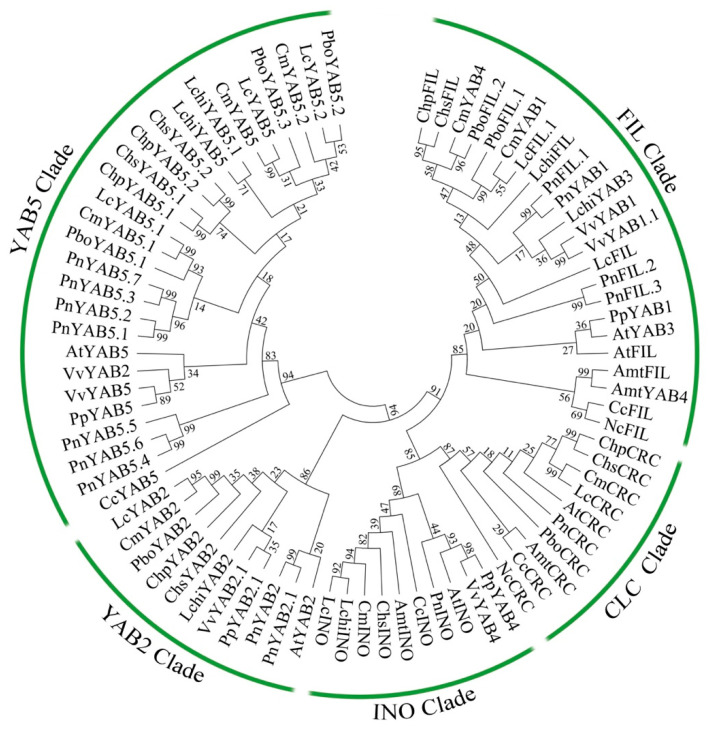
Phylogenetic tree of YABBY proteins from magnoliids and other angiosperms. The phylogenetic tree was constructed with the neighbor-joining (NJ) method using MEGA 7.0 software and was divided into five subgroups. The same acronym represents the same species: Cm: *C. micranthum*; Chp: *C. praecox*; Lc: *L. cubeba*; Chs: *C. salicifolius;* Pbo: *P. bournei*; Pn: *P. nigrum*; Lchi: *L. chinense*.

**Figure 4 plants-10-00021-f004:**
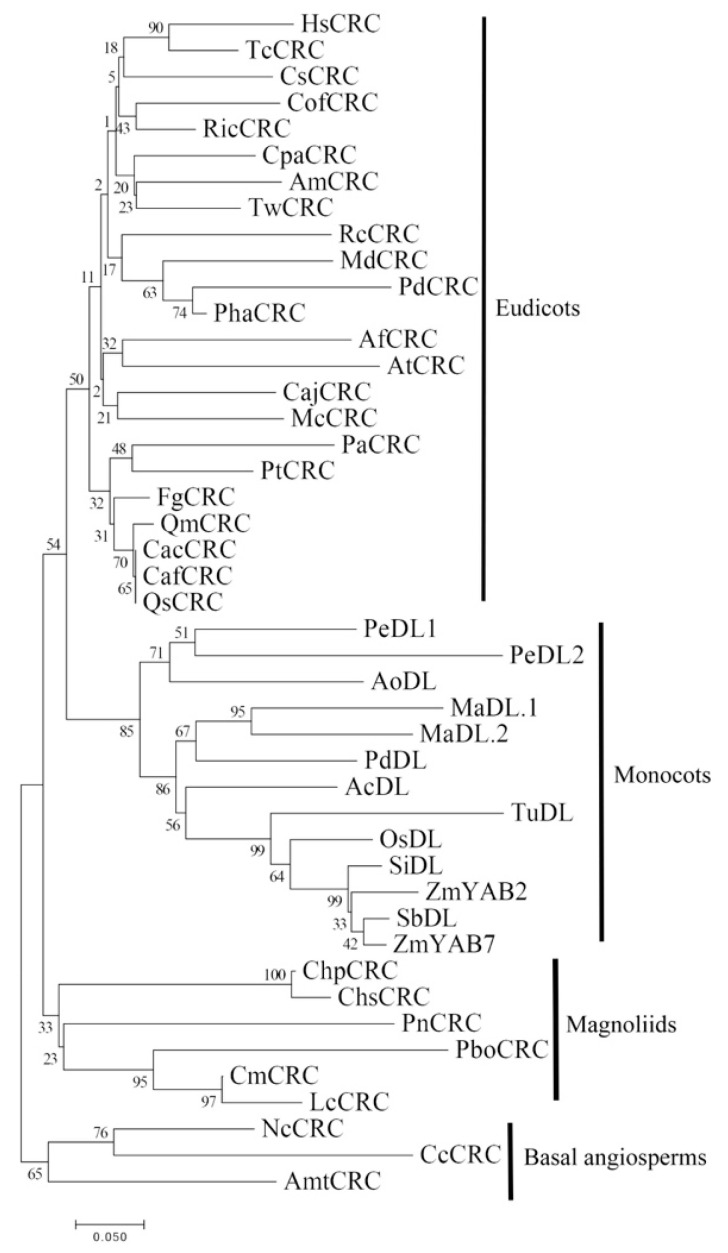
Phylogenetic tree of CRC/DL proteins from magnoliids and various angiosperms. The phylogenetic tree was constructed with the NJ method using MEGA 7.0 software. Bootstrap analysis was conducted with 1000 replications. The same acronym represents the same species: Cm: *C. micranthum*; Chp: *C. praecox*; Lc: *L. cubeba*; Chs: *C. salicifolius;* Pbo: *P. bournei*; Pn: *P. nigrum*; Lchi: *L. chinense*. The protein accession list for related proteins is collected in Appendix A.

**Figure 5 plants-10-00021-f005:**
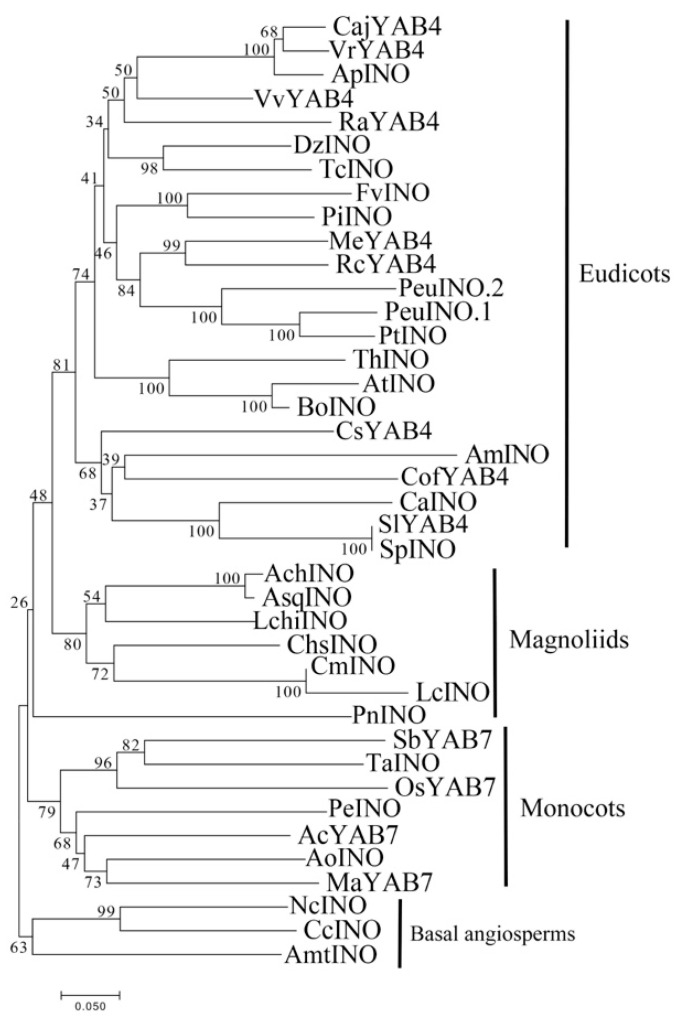
Phylogenetic tree of INO proteins from magnoliids and various angiosperms. The phylogenetic tree was constructed with the NJ method using MEGA 7.0 software. Bootstrap analysis was conducted with 1000 replications. The same acronym represents the same species: Cm: *C. micranthum*; Chp: *C. praecox*; Lc: *L. cubeba*; Chs: *C. salicifolius;* Pbo: *P. bournei*; Pn: *P. nigrum*; Lchi: *L. chinense*. The protein accession list for related proteins is collected in Appendix A.

**Figure 6 plants-10-00021-f006:**
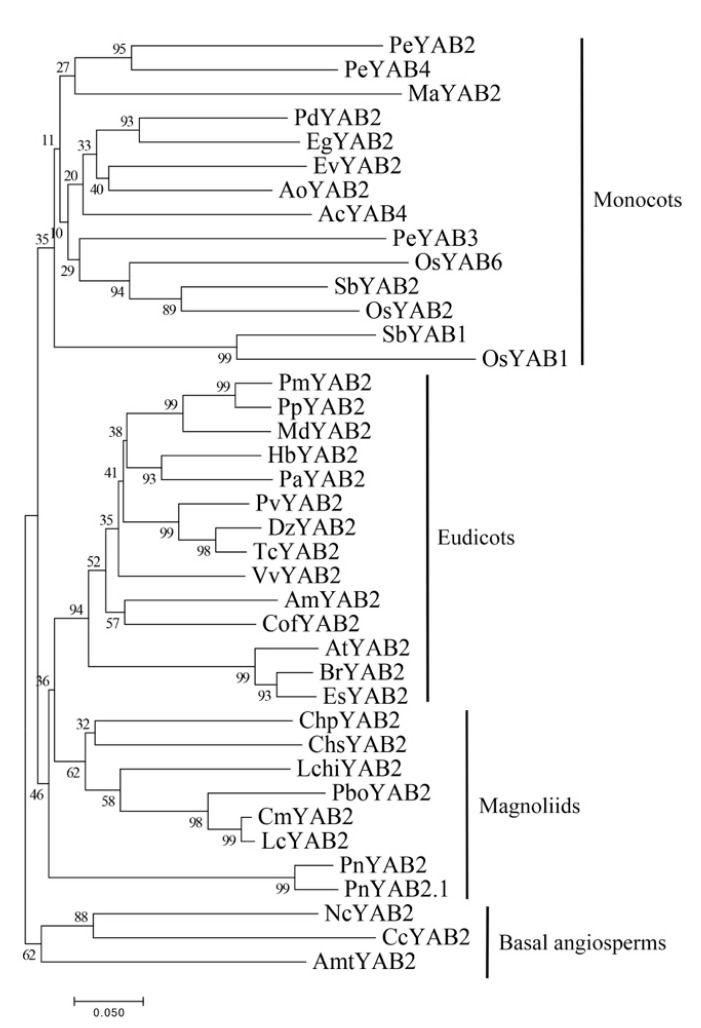
Phylogenetic tree of YAB2 proteins from various angiosperms. The phylogenetic tree was constructed with the NJ method using MEGA 7.0 software. Bootstrap analysis was conducted with 1000 replications. The same acronym represents the same species: Cm: *C. micranthum*; Chp: *C. praecox*; Lc: *L. cubeba*; Chs: *C. salicifolius;* Pbo: *P. bournei*; Pn: *P. nigrum*; Lchi: *L. chinense*. The protein accession list for related proteins is collected in Appendix A.

**Figure 7 plants-10-00021-f007:**
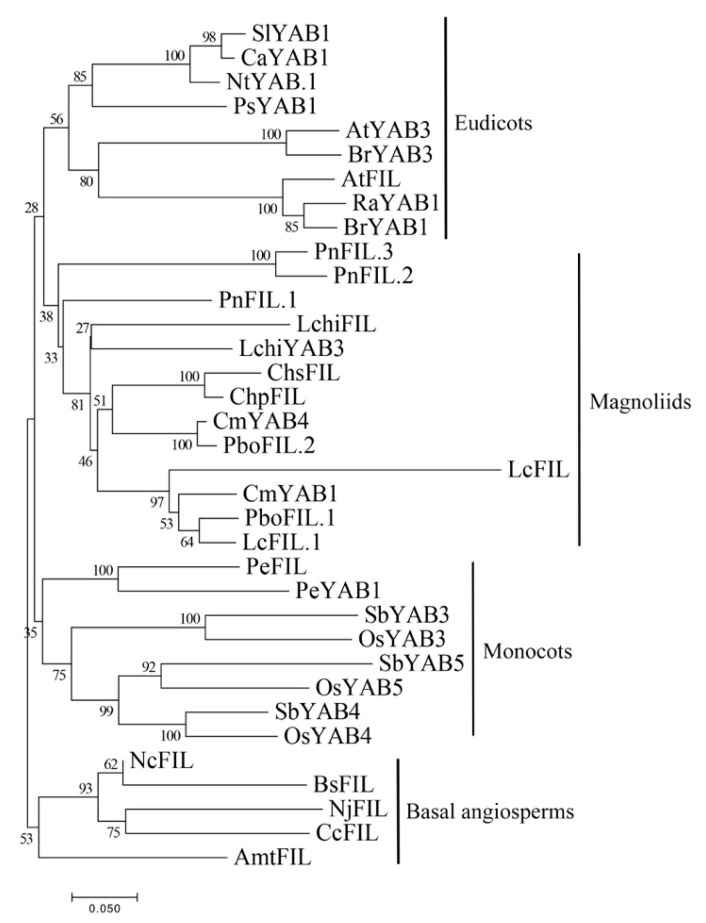
Phylogenetic tree of FIL proteins from the angiosperms. The phylogenetic tree was constructed with the NJ method using MEGA 7.0 software. Bootstrap analysis was conducted with 1000 replications. The same acronym represents the same species: Cm: *C. micranthum*; Chp: *C. praecox*; Lc: *L. cubeba*; Chs: *C. salicifolius;* Pbo: *P. bournei*; Pn: *P. nigrum*; Lchi: *L. chinense*. The protein accession list for related proteins is collected in Appendix A.

**Figure 8 plants-10-00021-f008:**
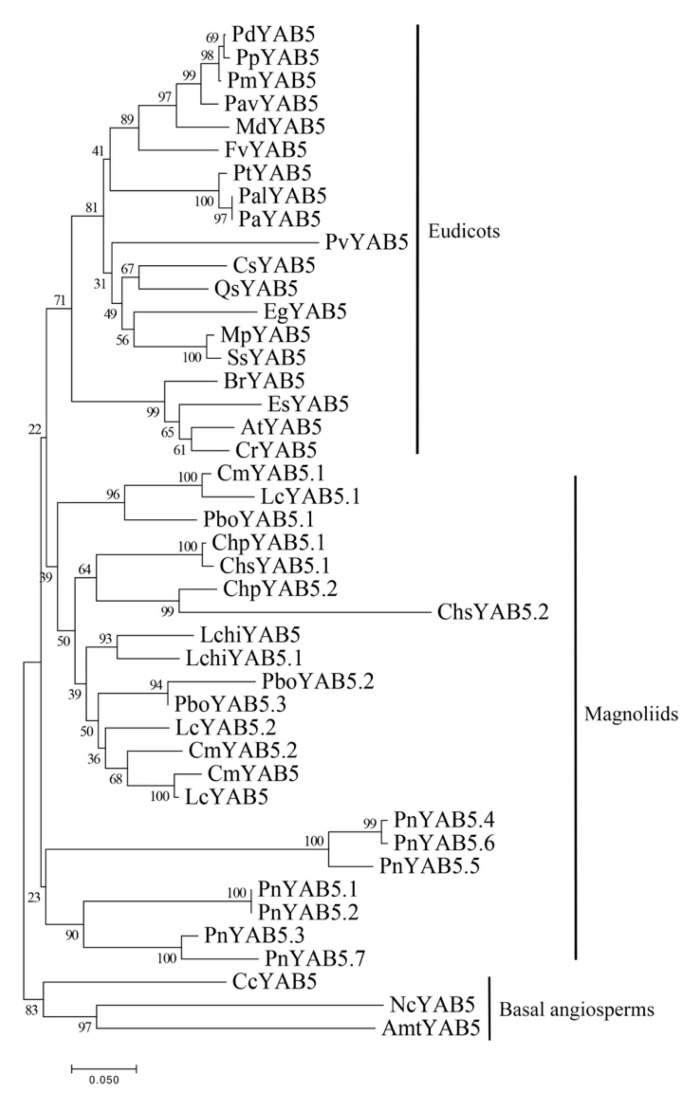
Phylogenetic tree of YAB5 proteins from various angiosperms. The phylogenetic tree was constructed with the NJ method using MEGA 7.0 software. Bootstrap analysis was conducted with 1000 replications. The same acronym represents the same species: Cm: *C. micranthum*; Chp: *C. praecox*; Lc: *L. cubeba*; Chs: *C. salicifolius;* Pbo: *P. bournei*; Pn: *P. nigrum*; Lchi: *L. chinense*. The protein accession list for related proteins is collected in Appendix A.

**Figure 9 plants-10-00021-f009:**
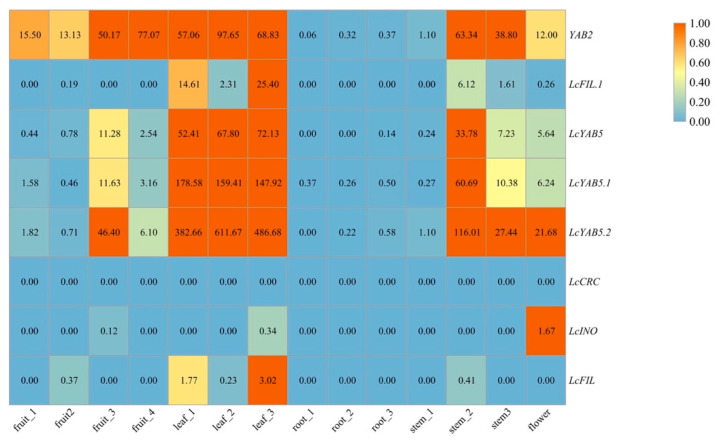
Gene expression profiles for the *YABBY* gene family in *Litsea*. The color scale above represents expression values in various organs, respectively.

**Table 1 plants-10-00021-t001:** The classification of the *YABBY* gene family in *Cinnamomum micranthum, Litsea cubeba, Phoebe bournei, Liriodendron chinense, Calycanthaceae praecox, Calycanthaceae salicifolius, and P. nigrum*.

Category	*C. micranthum*	*L. cubeba*	*P. bournei*	*L. chinense*	*C. praecox*	*C. salicifolius*	*P. nigrum*
CRC/DL-like	1	1	1	0	0	1	1
INO-like	1	1	0	1	1	1	1
YAB2-like	1	1	1	1	1	1	2
FIL-like	2	2	2	2	1	1	4
YAB5-like	3	3	3	2	2	2	7
Total	8	8	7	6	5	6	15

*CRABS CLAW, (CRC); DROOPING LEAF, (DL); INNER NO OUTER, (INO), YABBY2 (YAB2), FILAMENTOUS FLOWER (FIL), and YABBY5 (YAB5).*

**Table 2 plants-10-00021-t002:** The gene structures and physical properties of *YABBY* genes.

Clade	Species	Gene Name	ORF (bp)	Protein Length (aa)	pI	Mv (Da)	Exon 1	Exon 2	Exon 3	Exon 4	Exon 5	Exon 6	Exon 7	Exon 8	Exon 9	Exon 10	Exon 11	Exon 12	Exon 13	Exon 14	Exon 15
***CRC/DL***	*Chimonanthus salicifolius*	*ChsCRC*	534	177	7.05	20,169.74	47	68	75	48	117	119	74								
***CRC/DL***	*Cinnamomum micranthum*	*CmCRC*	423	140	9.04	15,858.01	68	128	90	48	75	8									
***CRC/DL***	*Litsea cubeba*	*LcCRC*	468	155	8.98	17,773.17	14	14	128	128	90	90	48	48	75	75	107	107			
***CRC/DL***	*Phoebe bournei*	*PboCRC*	1179	392	6.44	43,578.13	14	65	90	48	75	89	70	54	167	96	69	146	21	38	122
***CRC/DL***	*Piper nigrum*	*PnCRC*	519	172	9.40	19,208.75	71	131	90	48	75	62	35								
***CRC/DL***	*Chimonanthus praecox*	*ChpCRC*	519	172	8.82	19,406.98															
***FIL***	*Litsea cubeba*	*LcFIL*	534	177	9.87	19,463.36	40	76	47	155	47	114	47	114	173	202	77	88			
***FIL***	*Liriodendron chinense*	*LchiFIL*	615	204	8.37	22,893.1	68	74	75	48	75	114	101								
***FIL***	*Liriodendron chinense*	*LchiYAB3*	639	212	7.64	23,455.73	68	74	120	48	75	155	92								
***FIL***	*Piper nigrum*	*PnYAB1*	624	207	9.05	23,018.5	23	74	75	48	114	155	21	106							
***FIL***	*Cinnamomum micranthum*	*CmYAB4*	644	210	7.07	23,159.37	68	74	75	48	114	155	92								
***FIL***	*Phoebe bournei*	*PboFIL.1*	636	211	7.64	23,189.42	68	77	75	48	114	155	92								
***FIL***	*Phoebe bournei*	*PboFIL.2*	570	189	7.15	20,975.73	59	68	74	48	75	114	125								
***FIL***	*Cinnamomum micranthum*	*CmYAB1*	636	211	7.1	23,241.44	68	77	75	48	114	155	92								
***FIL***	*Piper nigrum*	*PnFIL.1*	636	211	6.78	23,373.68	68	74	75	48	114	155	95								
***FIL***	*Piper nigrum*	*PnFIL.3*	669	222	8.18	24,635.92	68	77	75	48	126	158	110								
***FIL***	*Piper nigrum*	*PnFIL.2*	669	222	8.19	24,655.98	68	77	75	48	126	158	110								
***FIL***	*Litsea cubeba*	*LcFIL.1*	717	238	7.07	26455.15	68	77	114	48	75	155	173								
***FIL***	*Chimonanthus salicifolius*	*ChsFIL*	663	119	7.62	12,999.94	68	74	75	48	117	155	64	54							
***FIL***	*Chimonanthus praecox*	*ChpFIL*	636	211	8.24	23,362.58															
***INO***	*Liriodendron chinense*	*LchiINO*	264	87	5.33	9615.4	86	176													
***INO***	*Liriodendron chinense*	*LchiINO*	264	87	5.33	9615.4	86	176													
***INO***	*Cinnamomum micranthum*	*CmINO*	546	181	6.38	20,548.4	38	134	99	51	75	92	50								
***INO***	*Chimonanthus salicifolius*	*ChsYAB4*	555	184	5.63	20,785.38	80	122	120	48	75	92	11								
***INO***	*Piper nigrum*	*PnINO*	615	204	4.88	22,845.96	98	134	120	48	75	35	98								
***YAB2***	*Liriodendron chinense*	*LchiYAB2*	495	164	8.8	18,231.7	23	116	114	48	75	113									
***YAB2***	*Chimonanthus praecox*	*ChpYAB2*	546	181	8.57	20,113.94															
***YAB2***	*Phoebe bournei*	*LcYAB2*	540	179	7.61	20,284.98	68	119	111	48	75	113									
***YAB2***	*Cinnamomum micranthum*	*CmYAB2*	540	179	7.03	20,329.98	68	119	111	48	75	113									
***YAB2***	*Chimonanthus salicifolius*	*ChsYAB2*	555	184	9.10	20,908.74	68	119	117	48	75	122									
***YAB2***	*Piper nigrum*	*PnYAB2*	573	190	8.79	21,523.58	68	122	126	48	75	128									
***YAB2***	*Piper nigrum*	*PnYAB2.1*	573	190	9.02	21,564.56	68	122	126	48	75	128									
***YAB2***	*Phoebe bournei*	*PboYAB2*	804	267	8.92	30,743.77	113	75	48	111	119	128	203								
***YAB5***	*Phoebe bournei*	*PboYAB5.3*	216	71	6.68	7689.93	74	140													
***YAB5***	*Phoebe bournei*	*PboYAB5.2*	387	129	5.41	14,506.45	65	125	119	74											
***YAB5***	*Phoebe bournei*	*PboYAB5.1*	477	158	8.45	17,567.2	74	150	48	75	65	59									
***YAB5***	*Litsea cubeba*	*LcYAB5.2*	483	160	8.53	18,198.95	74	134	91	75	65	38									
***YAB5***	*Litsea cubeba*	*LcYAB5*	507	168	8.2	18,764.34	74	119	117	48	75	68									
***YAB5***	*Litsea cubeba*	*LcYAB5.1*	813	270	9.46	30,701.78	25	138	119	117	48	75	65	37	180						
***YAB5***	*Cinnamomum micranthum*	*CmYAB5.2*	543	180	6.99	20,000.71	74	119	117	48	75	65	38								
***YAB5***	*Cinnamomum micranthum*	*CmYAB5.1*	771	183	9.26	20,592.77	74	149	117	48	75	65	17								
***YAB5***	*Cinnamomum micranthum*	*CmYAB5*	507	168	8.52	18,780.31	74	119	117	48	75	68									
***YAB5***	*Liriodendron chinense*	*LchiYAB5.1*	546	181	8.57	20,470.27	74	119	114	48	75	71	38								
***YAB5***	*Liriodendron chinense*	*LchiYAB5*	558	185	8.44	20,887.91	74	119	120	48	75	68	47								
***YAB5***	*Chimonanthus praecox*	*ChpYAB5.1*	555	184	6.99	20,693.56															
***YAB5***	*Chimonanthus praecox*	*ChpYAB5.2*	552	183	8.6	20,694.87															
***YAB5***	*Chimonanthus salicifolius*	*ChsYAB5.1*	555	184	6.99	20,724.53	74	119	117	48	75	68	47								
***YAB5***	*Chimonanthus salicifolius*	*ChsYAB5.2*	657	218	8.99	24,354.12	72	48	114	119	83	74	104	35							
***YAB5***	*Piper nigrum*	*PnYAB5.1*	513	170	8.80	19,384.9	74	122	126	48	75	62									
***YAB5***	*Piper nigrum*	*PnYAB5.2*	513	170	8.80	19,384.9	74	122	126	48	75	62									
***YAB5***	*Piper nigrum*	*PnYAB5.3*	507	168	8.85	19,035.58	74	122	120	48	75	62									
***YAB5***	*Piper nigrum*	*PnYAB5.4*	681	226	8.55	24,924.55	74	122	135	48	75	72	148								
***YAB5***	*Piper nigrum*	*PnYAB5.5*	576	191	8.37	21,440.51	74	68	75	48	135	122	47								
***YAB5***	*Piper nigrum*	*PnYAB5.6*	681	226	8.71	24,969.68	74	122	135	48	75	72	148								
***YAB5***	*Piper nigrum*	*PnYAB5.7*	510	169	8.89	19,289.98	68	119	117	48	75	122

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
