# Peer review of "Genome-Wide Identification of the YABBY Gene Family in Seven Species of Magnoliids and Expression Analysis in Litsea"

_plants, 2020, doi:10.3390/plants10010021_

Round 1

Reviewer 1 Report

The Manuscript focuses on genome-wide identification, as well as structural and phylogenetic characterization of the YABBY gene family in various species of magnoliids, as well as YABBY gene expression pattern analysis in one of these species. All studies were performed bioinformatically. Considering that YABBY transcription factors are plant-specific and very important evolutionary and developmental factors, the study is relevant.

The manuscript may be accepted for publication in Plants after a major revision.

First of all, all identified gene/protein sequences must be deposited in the NCBI GenBank. In the manuscript, authors should provide ID-numbers for each gene studied.

Please, include in all dendrograms (Fig. 3-8) the Arabidopsis thaliana YABBY proteins as the YABBYs from the model plant. In addition, in Figure 7, add another Arabidopsis FIL-like protein, AtYAB3.

L16-17: “However, until this study, nothing was known about the function of YABBY genes in magnoliids”

This is not entirely true. There are at least two publications:

  1. Jorge Lora 1, José I Hormaza, María Herrero, Charles S Gasser. Seedless fruits and the disruption of a conserved genetic pathway in angiosperm ovule development. Proc Natl Acad Sci U S A . 2011;108(13):5461-5. doi: 10.1073/pnas.1014514108.
  2. Zhineng Li, Yingjie Jiang, et al. Floral Scent Emission from Nectaries in the Adaxial Side of the Innermost and Middle Petals in Chimonanthus praecox. Int. J. Mol. Sci. 2018, 19(10), 3278; https://doi.org/10.3390/ijms19103278

Better to say: However, prior to this study, information on the function of the YABBY genes in magnoliids was extremely limited.

L27-28: what the authors mean by “temporal and perpetual level”? Apparently, temporal and spatial level.

L37-38: “it plays an important role in nutrition and reproductive tissue development” - it plays an important role in vegetative and reproductive tissue development.

L49: “and are both detected in leaf primordia and floral organs” - and are detected both in leaf primordia and floral organs

L50-51: “two tissue-specific expression genes, named CRC and INO genes, are restricted to developmental carpels and the outer integument” – two other genes, CRC and INO, are restricted to developing carpels and the outer integument of the ovule, respectively.

L56: “expression in ectopic growths of polarity mutants on the abaxial and adaxial surfaces” – expression in ectopic growths on the abaxial and adaxial surfaces of polarity mutants

L59-60: “GRAMINIFOLIA (GRAM) of FIL-like gene homolog, and PROLONGATA (PROL) of YAB5-like gene homolog” – FIL-like gene GRAMINIFOLIA (GRAM) and YAB5 homologous gene PROLONGATA (PROL)

L61: “regulate flower organ initiation and identity control” – regulate flower organ initiation and identity

L62: “motley” – varied

L65-66: “YAB2-like homolog gene from rice differentiated from previously reported, was” – YAB2 homologous gene from rice, was

L74: “Similarly, YABBY genes from Cabomba caroliniana (CcYAB5, CcFIL, CcINO, and CcCRC) are also expressed abaxially, as in Amborella and eudicots”

Should be replaced by “In contrast to Amborella, the YABBY genes from Cabomba caroliniana (CcYAB5, CcFIL, CcINO, and CcCRC) and other eudicots are expressed abaxially”.

L77-79: The first part of the sentence is clear, but the second is questionable. The authors write about adaxial-specific YABBY expression during the leaf primordia initiation stage in Zea. In references [1,4,19] I did not find a word about it; also [19] is focused on “The genome sequence of the orchid Phalaenopsis equestris”. How does this relate to the sentence?

Better to remove this phrase or rephrase it clearly and write exactly what the authors mean.

L79-82: "Nectarines" – nectaries; "walls" - whorl

It is necessary to rephrase, for example: “In eudicots, the expression of the CRC gene is restricted to nectaries and carpels; however, DROOPING LEAF (DL), an ortholog of CRC in monocots, is involved in the development of the leaf midrib and regulation of carpel identity by excluding B-class MADS-box gene expression from the flower fourth whorl.”

L 82-85: “On the other hand, INO cooperated with SUPERMAN (SUP) was involved in the asymmetric growth of the outer integument of ovules, which was another example of gene-specific functions for YABBY genes besides the differentiation of the CRC gene function”

It is necessary to rephrase, for example: The second reproductive YABBY gene, INO, in cooperation with SUPERMAN (SUP), is involved in the asymmetric growth of the outer ovule integument, which is another example of tissue-specific functions of the YABBY genes in addition to the CRC gene.

L85: “Arabidopsis INO expression was restricted”

…is restricted

L95-97: “… are not monocot, and with only one pollen pore, …”

…are not monocots, and with pollen having a single pore,…

L97-99: Again, there are two studies of YABBYs from magnoliids:

  1. Jorge Lora 1, José I Hormaza, María Herrero, Charles S Gasser. Seedless fruits and the disruption of a conserved genetic pathway in angiosperm ovule development. Proc Natl Acad Sci U S A. 2011; 108(13):5461-5. doi: 10.1073/pnas.1014514108.
  2. Zhineng Li, Yingjie Jiang, et al. Floral Scent Emission from Nectaries in the Adaxial Side of the Innermost and Middle Petals in Chimonanthus praecox. Int. J. Mol. Sci. 2018, 19(10), 3278; https://doi.org/10.3390/ijms19103278

Better to write: “However, little is known about the regulatory function of the YABBY genes in magnoliids, for example [Lora..2011, Li..2018], and the relationships… “

In the Discussion, the findings presented should be compared with those already published [1,2].

L110: “In previous reports, 8, 8, and 13 YABBY genes were identified in Arabidopsis, Oryza sativa,…”

In Arabidopsis, there are 6 YABBY genes, not 8.

L111-112: “In total, 55 and 56 YABBY genes were identified via HMM analysis and BlASTP search, respectively.”

Please, indicate where (in magnoliids) these genes were identified.

BlASTP - BLASTP

L140: “In contrast to the MADS-box gene family,”

This part of the sentence should be deleted as it does not apply to anything.

In the Figure 2 (MEME motifs), it is desirable to indicate on the right the names of the clades, to which the proteins belong.

L206-207: “CRC genes are positioned at the base of the clade containing monocot and eudicot CRC proteins, with the exception of the PnCRC gene,”

Here, the authors speak about proteins, therefore:

CRC proteins are located at the base of the clade containing monocot and eudicot CRC proteins, with the exception of the PnCRC protein,…

L230: all YAB2 proteins, not genes

Paragraph2.2: If the authors consider the phylogeny of magnoliids YABBY in such detail, it would be useful to provide a scheme of the evolutionary relationships of the species studied.

L274: “YAB5, and 7 to 12 in FIL clades (Table 2).”

Where is Table 2 or any illustrative Figure located?

L285-286: “YABBY genes are adaxially and conservatively expressed in seed plants [28].”

Abaxially!

Why is the reference [28] here – “The Phoebe genome sheds light on the evolution of magnoliids”??

L288: “tissues leaf” – leaf tissues

“that of LcFIL in this tissue (Figure 9),”

Where is the Figure 9 located??

L380: “evidence that magnoliids are diversified” – evidence that YABBYs of magnoliids are diversified

L386: “FIL genes are a variant in a different family” – The FIL genes are present in different numbers depending on the plant family

L392-393:Supplementary Table S1. Gene accession list of YABBY genes from different plants in phylogenetic tree.” - Supplementary Table S1. Accession numbers of the YABBY proteins from different plant species used in phylogenetic analysis.

In Supplementary file, the authors indicate that “AmtYAB5” has the ID NP_001292760.1.

According to the BLASTp against Arabidopsis thaliana, this protein is YAB2-like. Moreover, it is identical to AmtYAB2 – ID BAD72168.1 (twice included in the supplementary table).

Another protein, “AmtYAB2” (XP_011622458.1), according to BLASTp, had almost the same identity between YAB5 and YAB2 of Arabidopsis. Thus, it is difficult to say what exactly is a homologue.

Please, check the IDs and protein belonging to specific clades!

Furthermore, this table lists protein IDs, not gene accession numbers – please, correct it.

Considering all this, there are doubts about the clade-specificity of YAB5 and YAB2 genes identified by the authors in magnoliids. Again, each Figure should include Arabidopsis thaliana YABBY proteins.

Finally, species and gene names should be in italics: L110–128, 135, 181, 183, 202, 239-243, 258, 259, 287, 289, 322, 325, 327.

Protein names should be written in non-italics: L186-194, 364.

Author Response

We sincerely thank the editor and reviewers for their constructive comments and suggestions. These comments were helpful for improving our paper. We have incorporated the suggestions and completed the requested revision of our manuscript. 

General response: All identified gene/protein sequences has been deposited in the NCBI GenBank. The Arabidopsis thaliana YABBY proteins has been included in all dendrograms (Fig. 3-8) in the manuscript.

The main modifications of the manuscript are listed as in responses file with point-by-point.

Reviewer 2 Report

  1. Abstract: lines 25 & 26 - This sentence is redundant, must be deleted.
  2. Introduction: line 69 - "to be unrelated to" must be followed by a noun, not a verb. line 94 - would "unique" be more correct than "typical?" line 95 - "two seed leaves" should be "two cotyledons."  line 102 and throughout the text - species names should be italicized. 
  3. Results: lines 109 and 175 - There are two "2.1." sub-sections. line 139 - Is "Congruence of YABBY and zinc finger domains" a sub-section?
  4. If authors can provide a figure showing the physical map positions of the YABBY gene family on chromosomes of Arabidopsis, Oryza sativaZea mays, and the species of magnoliids, the value of this manuscript will be greatly enhanced.

Author Response

We sincerely thank the editor and reviewers for their constructive comments and suggestions. These comments were helpful for improving our paper. We have incorporated the suggestions and completed the requested revision of our manuscript.

General response: The figure showing the physical map positions of the YABBY gene family on chromosomes of ArabidopsisOryza sativaZea mays, and the six species of magnoliids with the exception of Chimonanthus praecox, has been provided in Supplementary Figure S1-2. 

The main modifications of the manuscript are listed in response file with point-by-point.

Round 2

Reviewer 1 Report

I am satisfied with the corrections in the text of the article and the work can be published.

Nonetheless, please, check again all genes and species names - not all of them are in italics.

Author Response

Dear Editors, We sincerely thank the editor and reviewers for their constructive comments and suggestions. The comments were helpful for improving our paper. We have incorporated the suggestions and completed the requested revision of our manuscript. We hope that the revised manuscript meets the journal requirements for publication.
